# The Running Injury Continuum: A qualitative examination of recreational runners' description and management of injury

Aisling Lacey [1,2]*, Enda Whyte[1,3], Sinéad O'Keeffe[1,3], Siobhán O'Connor[1,3], Aoife Burke[1,3], Kieran Moran[1,2,3]

1 School of Health and Human Performance, Dublin City University, Dublin, Ireland, 2 Insight SFI Research Centre for Data Analytics, Dublin, Ireland, 3 Centre for Injury Prevention and Performance, School of Health and Human Performance, Dublin City University, Dublin, Ireland

* aisling.lacey5@mail.dcu.ie

## Abstract

### Introduction

A critical step in understanding and preventing running-related injuries (RRIs) is appropriately defining RRIs. Current definitions of RRIs may not represent the full process of injury development, failing to capture lower levels of injury that many athletes continue to train through. Understanding runners' description and management of the injury development process may allow for a more appropriate examination of all levels of injury. This study aimed to examine recreational runners' description and management of the injury development process.

### Methods

A qualitative focus group study was undertaken. Seven semi-structured focus groups with male (n = 13) and female (n = 18) recreational runners took place. Focus groups were audio and video recorded, and transcribed verbatim. Transcripts were reflexively thematically analysed. A critical friend approach was taken to data coding. Multiple methods of trustworthiness were executed.

### Results

Runners describe injury on a nine-level continuum, ranging from injury-free to career-ending injury. There are lower and higher levels of injury. Each level of injury is described across four categories of descriptors; physical description, outcome (effect on running and daily life), psychological description, and management.

### Conclusion

The Running Injury Continuum is a tool that can be used for injury surveillance (for healthcare professionals and researchers) and for research investigating RRI risk factors. Healthcare professionals, researchers and coaches must ensure they monitor the development of all levels of RRIs, across all categories of descriptors. Runners need to be educated

**Data Availability Statement:** All relevant data are within the manuscript and its Supporting information files.

**Funding:** KM received funding for this research article from Science Foundation Ireland (SFI) under grant number SFI/12/RC/2289_P2, co-founded by the European Regional Development Fund. Fundings for the study was received as part of a larger-scale, centre-wide funding from Science Foundation Ireland to develop Insight (the national research centre for data analytics: www.insight-centre.org). The funders had no role in study design, data collection and analysis, decision to publish or preparation of the manuscript.

**Competing interests:** The authors have declared that no competing interests exist.

regarding appropriate self-management strategies for lower level injuries, with access to evidence-based information being a critical management tool.

## Introduction

Despite recreational running being an extremely popular physical activity and sport [1] it is associated with high rates of injury [2], with incidence rates of 40% [3] or 7.7 injuries/1000 hours [4] reported. The vast majority of running-related injuries (RRIs) are overuse injuries [5] which occur when excessive, repetitive loads are applied to tissues beyond their adaptive capability [5–7]. While RRIs develop from the interaction between multiple risk factors [5, 8, 9], there is little consensus on what these risk factors are [10, 11]. This is potentially due to a commonly employed approach that uses definitions of injury that are limited, focusing on injuries causing time-loss, rather than capturing the process of injury development [12–14].

Clearly, a critical step in understanding and ultimately preventing RRIs is appropriately defining a RRI. Traditionally, time-loss from activity has been the main criterion for defining overuse injuries, with the duration of time-loss determining severity [15]. The current consensus definition of a RRI expands on this definition to also include training restriction or the need for medical attention as criteria [16]. Although this approach broadens the scope of a RRI, it may not represent the full process of injury development, failing to capture the *lower levels of injury* that many athletes continue to train through [15]. It has been acknowledged that *lower levels of injury* exist prior to those that cause time-loss, as evident from: (i) a general physiological perspective as represented in the Well-Being Continuum [17], (ii) a general sports perspective through the development of the Oslo Sports Trauma Research Centre Overuse Injury Questionnaire (OSTRC-Q) [15, 18], and (iii) directly from capturing runners' perceptions of RRIs through the Injury Pathway model [19].

Understanding runners' perceptions of RRIs may allow for a more appropriate examination of these *lower levels of injury*. Only five studies appear to have reported on runners' perception of the process of injury development [19–23], with just two studies alluding to runners' perception of *lower levels of injury* [19, 22]. These studies reported that RRIs are perceived as progressive, with injury 'categories' suggested in one study, although not described further [22], and the identification of a 'complaint' level in the process of injury development (prior to time-loss, training restriction or seeking medical attention) described in another [19]. However, it is unclear if this 'complaint' level of injury is a single level, or if it comprises multiple unique levels of injury. Therefore, a greater understanding of this process of injury development (i.e., runners' description and management of each level of this process) is clearly needed if researchers and clinicians are to better understand the multifactorial and progressive nature of RRIs, their risk factors, and ultimate prevention.

It is also important for future research aiming to identify RRI risk factors to understand how *lower levels of injury* may interact with other risk factors to develop into a significant injury (i.e., as per the consensus definition [16]) [17], or indeed, how these *lower levels of injury* may themselves be risk factors for injury [17]. In addition, with the dynamic relationship between multiple risk factors and the onset of injury, it is important to understand how runners react to this process of injury development (i.e. a *lower level injury*), and how they manage all levels of injury. This may not only help to differentiate between various levels of injury, but also provide insight into how these levels act as potential risk factors themselves for further injury. Not considering these *lower levels of injury* could potentially mask important information that is relevant in identifying risk factors for injury. Therefore, the aim of the present

study was to explore recreational runners' description of injury, and their management of the process of injury development.

## Methods

### Design

Interpretative phenomenology (IP) was deemed to be an appropriate methodological approach because, as a branch of phenomenology, it focuses on the lived experiences of humans, eliciting insightful accounts of individuals' subjective experiences regarding a certain topic [24, 25]. Focus groups were deemed an appropriate method of data collection as they can yield rich, in-depth data through the interaction of participants [26–28], and can enhance personal accounts by benefitting from the rapport built in a homogeneous sample [29]. Although suggested to be incompatible by some [30], executing focus groups with an interpretative phenomenological approach (IPA) was congruent with the aim of the study and allowed for enrichment of the data regarding the phenomenon of interest through the interaction between participants [31]. Despite the complex interactional environment that is created by conducting focus groups, the opportunity to engage with a homogenous sample (i.e., runners) as a group can elicit insightful and experiential data [32]. While it has been suggested that there is difficulty in developing phenomenological accounts of data due to the complexity of group dynamics (from interactional, social and contextual perspectives) [33], the complexity of individual and shared contexts [33], and the influence and position of the researcher(s) [33–35], the free-flowing and engaging nature of focus group discussion can allow for social interaction of shared, similar or conflicting lived experiences of the phenomenon of interest [31, 36].

### Participants

A purposive sample of 31 adult recreational runners were recruited. Between April and June 2022 local running clubs were contacted via email or telephone and asked to distribute research information and contact details of researchers to potential participants. Those interested then contacted the researchers. Eligible participants were recreational runners (someone running at least once per week for the previous six months [37], aged 18 years or older, and had no previous education or training in musculoskeletal injuries (e.g., Athletic Therapist or Physiotherapist). Participant demographics are presented below.

### Pilot study

A semi-structured focus group schedule was developed during several brainstorming meetings between researchers (AL, EW, SOK, and KM [S1 Table]). Question content, sequencing, phrasing and timing were discussed during meetings. The schedule was tested on colleagues to determine its appropriateness, and then used in the pilot study (details below). A pilot study was conducted to test the focus group schedule and develop an 'order of themes' document, which was used to organise the coding of focus groups. Five male and five female physically active participants were recruited as a convenience sample, aged 23.8 ± 5.9 years. Two focus groups were moderated by two researchers (AL and SOK), each taking place in-person, on University grounds, and lasted 59.7 ± 5.6 minutes. Focus groups were audio and video recorded, and transcribed verbatim. Data obtained from the pilot study are not included as part of the results.

## Main study procedures

Data was collected between April and June 2022. Ethical approval was granted by the local university's Ethics Committee (DCUREC/2022/071), and the Standards for Reporting Qualitative Research were adhered to [38] (S2 Table). Participants were organised into groups based on their availability to attend, running background and age (brief demographic information was collected prior to each focus group). Seven focus groups were moderated by two researchers (AL and SOK) and lasted 83.9 ± 18.1 minutes. On arrival to each focus group, participants were introduced to one another by the moderators, and a casual conversation (not recorded) took place prior to starting. Authors (AL and SOK) had access to participants' identifying information during data collection. Participants provided informed consent before commencing. Each focus group began with a brief introduction and the aims of the study were outlined (S1 Table). Participants were encouraged to speak freely, ask each other questions, and were given the opportunity to ask the moderators questions at any point. Firstly, participants were asked how they would define a RRI, and then how they would describe a RRI (approx. 10 minutes of discussion). Participants were prompted to elaborate on their descriptions and asked to draw (as a group) their description of RRIs on a whiteboard (approx. 45 minutes of discussion). They were then asked how they manage RRIs, and asked to insert these descriptions on their drawing (approx. 30 mins). Conversation pursued naturally, not being led by the moderators, but with the moderators prompting participants to give as much detail as possible. Participants were asked to complete a short individual questionnaire (hard-copy) gathering further demographic information, training practices, and injury history (including their experience of running-related pain/discomfort). Questions included were in the form of Likert scales and open-ended responses. Additionally on this questionnaire, participants were asked to draw their individual perception of how a RRI progresses (approx. 5 mins). On closing the focus groups, participants were given another opportunity to ask questions or provide additional information. The progressive nature of RRIs was not described to participants, nor was the concept of *levels of injuries*.

A reflective and iterative approach was taken to focus groups moderation. Following each focus group, moderators discussed their perception of each focus group, expressed their opinions on the appropriateness of the focus group schedule, and discussed how they could potentially improve for the next group. During the main data collection phase, both moderators included additional probes to encourage further explanation of some points raised and to encourage all participants to share their perceptions.

## Data analysis

Frequencies and descriptive statistics were generated from the questionnaire responses using SPSS (IBM Corporation; version 27) and all participants were given an identification number and coded by self-identified gender (e.g., male 4 = M4, female 2 = F2) in order to maintain anonymity. All focus groups were audio and video recorded. Audio recordings were sent to an external transcription service to be transcribed verbatim. On their return, the primary author reviewed the transcripts alongside the video recordings, corrected any discrepancies, inserted nuance (to account for sarcasm and gestures), and assigned dialogue to the according speaker.

The transcribed focus groups were coded by the primary author using NVivo (QSR International; release 1.6.2). A reflexive thematic analysis approach was taken to data analysis according to Braun and Clarke's principles [39, 40]. This process followed six recursive phases: (i) the primary author familiarised herself with the data by reading the transcripts, correcting discrepancies, adding nuance, and re-reading the transcripts, (ii) brief labels (codes) were generated to identify important aspects of the data, (iii) themes were generated through examining

and organizing the codes, (iv) themes were then reviewed against the whole dataset, and developed further, (v) developed themes were then refined, defined and named, and (vi) the data were organised into a written report [39, 40].

The 'order of themes' document was used to organise the codes, sub-themes, themes and core categories and was reflexively updated throughout data collection and analysis phases. Based on the developing coding, further levels of sub-themes were developed while some sub-themes were merged. Additionally, as a level of data triangulation, the primary author reviewed each group's whiteboard drawing alongside the corresponding transcript and video recording. Further detail from the transcript was added to each drawing, which clarified these visual representations and ensured consistency between transcripts and drawings. All seven drawings were combined to develop the final nine-point continuum. The final Running Injury Continuum was arrived at based on how participants described each level and where they placed each level. Constant comparative analysis took place, beginning after transcription of the first focus group and continued throughout data collection [41]. A visualisation of the methods of data analysis is included in the supplementary material (S1 Fig). A similar sample size has been used in previous research and allowed for the collection of a rich description from participants regarding RRIs [42, 43], as well as aligning with the aims of the study.

## Trustworthiness

Throughout the data collection and analysis phases, regular discussions on the developing codes, sub-themes, themes and core categories ensued (between AL, EW, SOK and KM), which challenged and facilitated multiple interpretations of the data. To further enhance the analytical process [44] and to ensure reliability and rigour of results presented [45], a critical friend approach was taken between researchers. The critical friend approach encourages reflexivity in the co-construction of knowledge [45, 46], and facilitates the exploration of multiple interpretations of the data, reducing the potential for researcher bias [44, 47]. Firstly, researchers (AL and SOK) met on multiple occasions to reflexively discuss and review the 'order of themes' document. After all focus groups had been coded by the primary author, a percentage (approximately 30%) of the transcripts were coded by another experienced qualitative researcher (SOK). Taking a critical friends approach again, researchers (AL and SOK) met on multiple occasions to discuss their interpretations of the transcripts, challenging each other's interpretations of the data. Codes, sub-themes, themes and core categories were critically reviewed and discussed. The process facilitated the development of additional codes, while some existing sub-themes were merged/expanded.

Trustworthiness was further enhanced via investigator triangulation. Researchers (AL, SOK, EW and KM) met on several occasions throughout the data collection and analysis phases to review the coding of the data and discuss individual interpretations. On finishing data collection and analysis, a wider group of researchers (AL, EW, SOK, SOC, AB and KM) met to discuss their interpretations of the findings. Multiple interpretations of the data were presented and discussed during meetings, coding was further refined, and a thorough representation of participants' experiences and perceptions of RRIs was presented.

Additionally, all individual transcripts and whiteboard drawings were returned to participants within four weeks of their focus group, giving participants the opportunity to alter their individual dialogue and/or group drawing. We also ensured that each participant was satisfied with our interpretation of their discussion. No requests were received to alter group drawings or individual dialogue. Furthermore, multiple examples of direct quotations from participants are presented, enhancing the accuracy and trustworthiness of findings. A broad and diverse contribution from participants is also included, reducing the likelihood of individual bias [48].

# Results

## Demographics

Seven focus groups were conducted with 18 (58%) female and 13 (42%) male recreational runners. Participants were aged 39.7 ± 12.7 years (range 20–65 years). The majority of runners had been running for more than 5 years (n = 17, 55%), fewer running 1–3 years (n = 10, 32%), and the least amount running 4–5 years (n = 4, 13%). Participants trained either 2–3 times per week (n = 16, 52%) or 4–6 times per week (n = 15, 48%), with the majority of runners participating in organised running events (n = 27, 87%). Most (n = 17, 55%) runners predominately trained on their own, with the remainder training in small (n = 7, 23%) or large (n = 7, 23%) groups. Injury prevention was 'very' or 'extremely' important to the majority of runners (n = 21, 68%), with fewer stating that it was 'slightly' or 'moderately' important (n = 9, 29%), and just one participant reporting it was 'not at all' important (n = 1, 3%). Further details on training practices are presented in Table 1. According to the consensus definition of injury [16], most runners reported having a previous RRI (n = 27, 87%), while few reported never having a RRI (n = 4, 13%). Nineteen participants had at least one RRI in the previous 12 months (n = 19, 70%). All participants reported previously experiencing a *lower level injury*. Further details on RRI history are detailed in Table 1.

## Runners' description of injury

Runners described RRIs and the process of injury development on a nine-level continuum, with each level increasing in severity of injury (Fig 1). The nine levels of RRIs identified were: 'running smooth' (no injury), 'discomfort', 'niggle', 'twinge', 'persisting niggle', 'non-responding niggle', 'short-term injury', 'long-term injury', and 'career-ending injury' (Table 2).

**Table 1. Running practices and injury history.**

| | |
|---|---|
| **Preferred running events (n = 31)*** | *<5km*: 16% (n = 5) |
| | *5km*: 67% (n = 21) |
| | *10km*: 58% (n = 18) |
| | *16km*: 13% (n = 4) |
| | *Half marathon (21.1km)*: 19% (n = 6) |
| | *Marathon (42.2km)*: 10% (n = 3) |
| | *Ultramarathon (>42.2km)*: 3% (n = 1) |
| | *Triathlon*: 3% (n = 1) |
| | *Other*: 3% (n = 1) |
| **Weekly mileage (n = 31)** | *<10km*: 3% (n = 1) |
| | *10-20km*: 39% (n = 12) |
| | *21-30km*: 32% (n = 10) |
| | *31-40km*: 3% (n = 1) |
| | *41-50km*: 16% (n = 5) |
| | *>50km*: 7% (n = 2) |
| **Amount of missed training with worse ever RRI (n = 27)** | *<1 week*: 7% (n = 2) |
| | *7–10 days*: 15% (n = 4) |
| | *2–3 weeks*: 19% (n = 5) |
| | *4–6 weeks*: 7% (n = 2) |
| | *>6 weeks*: 48% (n = 13) |

*: multiple choice available, RRI: running-related injury.

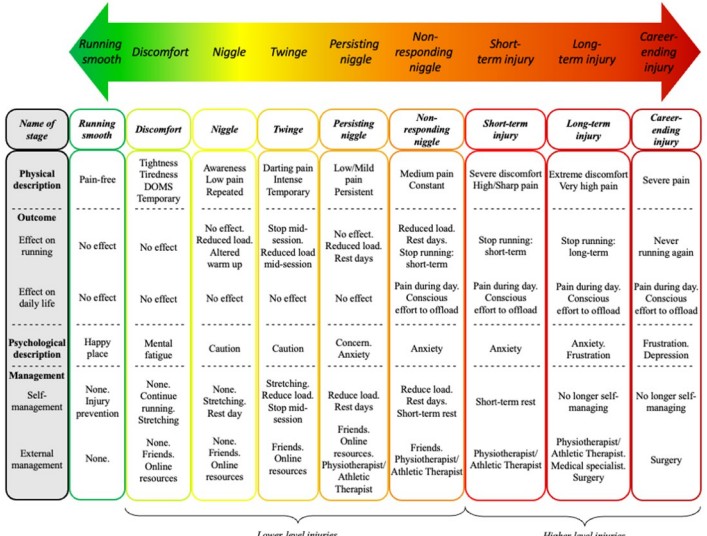

**Fig 1. The Running Injury Continuum.** Description and management of each level of the Running Injury Continuum. DOMS: Delayed Onset Muscle Soreness.

Runners described each level of the Running Injury Continuum in terms of four key categories of descriptors: physical description, outcome (effect on running and effect on daily life), psychological description and management (Fig 1, Table 2).

## The Running Injury Continuum

**Running smooth.** 'Running smooth' was described as *"zero pain"* (M4), *"smooth"* (M3) and *"completely perfect"* (M9). Runners did not describe a negative physical complaint, a negative outcome, or a negative psychological reaction with this level.

In relation to managing this first level, some participants suggested they would complete injury prevention exercises because they *"don't want injury to happen"* (F11), so they *"try to prevent it by doing* [their] *stretching"* (F11). However, some runners also suggested that no management strategies would be taken at this level, suggesting that they would be *"complacent"* (F10) when injury free, and they would *"wait for something to happen and then treat it"*(F10), being *"reactive rather than proactive"* (F10) to injury management.

**Discomfort.** 'Discomfort' was described with terms and phrases such as *"tightness"* (F17), *"tiredness"* (M3), *"a little bit of pain"* (F6), and *"stiffness"* (F3). Many participants also suggested that this level could be associated with previous training (i.e., delayed onset muscle soreness [DOMS]) and *"being tired from running the previous day"* (M3). They described it as something *"temporary"* (M8) and that they *"know that* [it] *will go away"* (F16);

> *"I think discomfort is fairly mild, no injury. Discomfort is something that is there. It isn't going to stop you... You could have discomfort on one run and go out two days later and it is gone"*
>
> (F11)

Some participants described this level of injury as a *"satisfying pain"* (F6) and perceived that if they experience discomfort, it means they have worked hard and completed a 'good session',

**Table 2. 'Order of themes' document: Description and management of each level of the Running Injury Continuum.**

|   | Core category | Theme | Sub-theme | Secondary sub-theme | Tertiary sub-theme |
|---|---|---|---|---|---|
| **1** | **Running Smooth** | | | | |
| | Physical description | Sensation | Pain free | | |
| | Management | Injury prevention | | | |
| | | No management required | | | |
| | Psychological description | Happy place | | | |
| **2** | **Discomfort** | | | | |
| | Physical description | Sensation | Tightness | | |
| | | | Delayed onset muscle soreness (DOMS) | | |
| | | | Tiredness | | |
| | | | Low pain | | |
| | | | Stiffness | | |
| | | | Uncomfortable | | |
| | | Frequency/Onset | Temporary | | |
| | | | Infrequent | | |
| | | Precursor to injury | | | |
| | Outcome | Effect on running | Full training | Stick to training plan | |
| | | Effect on daily life | | | |
| | Psychological description | Mental fatigue | | | |
| | Management | Self-management | Additional stretching | | |
| | | | Continue training | | |
| | | | Therapies | Massage gun | |
| | | External management | Friends | | |
| | | | Internet sources | Google | |
| | | No management | | | |

(*Continued*)

**Table 2.** (Continued)

| | Core category | Theme | Sub-theme | Secondary sub-theme | Tertiary sub-theme |
|---|---|---|---|---|---|
| **3** | **Niggle** | | | | |
| | Physical description | Sensation | Pain | Severity of pain | Low pain |
| | | | | | Not painful |
| | | | | | Dull pain |
| | | | | Type of pain | Aches |
| | | | Awareness | Different from opposite side | |
| | | | Discomfort | | |
| | | | Tightness | | |
| | | | Irritation | | |
| | | | Tiredness | | |
| | | Frequency/Onset | Repeated | | |
| | | | Temporary | | |
| | | | Constant | | |
| | | | Post-session complaint | | |
| | | | Felt while running | | |
| | | Precursor to injury | From an unknown cause | | |
| | Outcome | Effect on running | Full training | Run through it | |
| | | | | Pressure to continue | Desire to continue |
| | | | | | Stick to training plan |
| | | | | Doesn't affect training | |
| | | | | Can ignore | |
| | | | Altered training | Altered warm up | |
| | | | | Forced change to training | Reduce load |
| | | | | Altered technique | |
| | | Effect on daily life | No effect | | |
| | Psychological description | Caution | | | |
| | | Affects motivation to train | | | |
| | | Annoying | | | |
| | Management | Self-management | Altered training | Additional stretching | |
| | | | | Rest | Additional rest day |
| | | | | Reduce load | |
| | | | | Session preparation | |
| | | | | Change technique | |
| | | | Therapies | Foam rolling | |
| | | | Strength & conditioning | | |
| | | | Accessory supports | Footwear | |
| | | External management | Friends | | |
| | | | Internet sources | YouTube exercises | |
| | | | Massage | | |
| | | No management | Resolves on its own | | |
| | | | Ignore niggle | | |
| | | | Ignore advice | | |

(*Continued*)

**Table 2.** (Continued)

| | Core category | Theme | Sub-theme | Secondary sub-theme | Tertiary sub-theme |
|---|---|---|---|---|---|
| **4** | **Twinge** | | | | |
| | Physical description | Sensation | Pain | Type of pain | Darting pain |
| | | Frequency/Onset | Temporary (short-lived) | | |
| | Outcome | Effect on running | Altered training | Stop mid-session | |
| | | | | Reduced load | |
| | | | Full training | Run through it | |
| | Psychological description | Caution | | | |
| | | Annoying | | | |
| | Management | Self-management | Altered training | Reduce load | |
| | | | | Rest | Stop mid-session |
| | | | | Additional stretching | |
| | | External management | Friends | | |
| | | | Google | | |
| **5** | **Persisting niggle** | | | | |
| | Physical description | Frequency/Onset | Persistent | | |
| | | Sensation | Pain | Severity of pain | |
| | Outcome | Effect on performance | Altered training | Reduce load | |
| | | | | Rest days | |
| | Psychological description | Anxiety | | | |
| | | Annoyed | | | |
| | Management | Self-management | Altered training | Reduce load | |
| | | | | Stop running | Short-term rest |
| | | | Therapies | | |
| | | External management | Friends | | |
| | | | Athletic Therapist/Physiotherapist | | |
| | | | Internet sources | Google | |
| **6** | **Non-responding niggle** | | | | |
| | Physical description | Sensation | Pain | Pain stops running | |
| | | Frequency/Onset | Constant | | |
| | | | Pain while running | | |
| | Outcome | Effect on running | Altered training | Reduce load | |
| | | | | Altered technique | |
| | | | Continue to train | | |
| | | | Stop running | | |
| | | Effect on daily life | Effects daily life | Pain during day | |
| | | | | Conscious off-loading | |
| | Psychological description | Anxiety | | | |
| | Management | Self-management | Altered training | Stop running | Rest |
| | | | | Additional stretching | |
| | | | | Reduce load | |
| | | | Medication | | |
| | | External management | Athletic Therapist/Physiotherapist | | |
| | | | Friends | | |
| | | | Failed self-management | | |

(*Continued*)

**Table 2.** (Continued)

| | Core category | Theme | Sub-theme | Secondary sub-theme | Tertiary sub-theme |
|---|---|---|---|---|---|
| **7** | **Short-term injury** | | | | |
| | Physical description | Frequency/Onset | Progressively worsening | | |
| | | | Acute onset | 'Snap' | |
| | | | | 'Pulled muscle' | |
| | | Sensation | Pain | Type of pain | Severe discomfort |
| | Outcome | Effect on running | Stop running | | |
| | | | Continue to run (with pain) | | |
| | | Effect on daily life | Effects daily life | Pain during day | |
| | | | | Conscious off-loading | |
| | Psychological description | Anxiety | | | |
| | Management | External management | Athletic Therapist/Physiotherapist | Stop running | |
| | | Self-management | Rest | | |
| **8** | **Long-term injury** | | | | |
| | Physical description | Sensation | Pain | Extreme discomfort | |
| | | | | Severe pain | |
| | Outcome | Effect on running | Stop running | Unable to run–long term (months) | |
| | | Effect on daily life | Effects daily life | Pain during day | |
| | | | | Conscious off-loading | |
| | Psychological description | Anxiety | | | |
| | | Frustration | | | |
| | Management | External management | Medical speciality | | |
| | | | Athletic Therapist/Physiotherapist | | |
| | | Self-management | Stop running | | |
| **9** | **Career-ending injury** | | | | |
| | Physical description | Frequency/Onset | Constant | | |
| | | | Pain outside running | | |
| | | Sensation | Pain | Severity of pain | Very high |
| | Outcome | Effect on running | Unable to run–permanently | | |
| | | | Career-ending injury | | |
| | | Effect on daily life | Effects daily life | Pain during day | |
| | | | | Conscious off-loading | |
| | Psychological description | Frustration | | | |
| | | Depression | | | |
| | Management | External management | Requires surgery | | |

but they are also confident that it will not persist into a more serious injury; *"There is a good tightness. Like your quads after a speed session"* (F18). However, some runners described this level as a *"precursor to injury"* (M1) and suggested that further injury can develop from this level because *"you have planted the little injury seed. . . the injury is on its way"* (M4). With regard to the outcome, this level was not suggested to cause a negative effect on running, with participants suggesting that *"you'd definitely run"* (F6) and complete full training at this level; *"you don't back off your mileage because you are tight"* (F15). Additionally, this level did not have a negative effect on a runner's daily life, however, some runners associated it with 'mental fatigue' and described how they *"don't feel mentally strong"* at this level (M3).

The majority of runners take no management strategies because *"you know you will recover in half a day, in two days, three days"*(M8) and continuing training is the best management

strategy; *"for the stiffness one, the treatment of that one probably would be to go for a run. . . to loosen it out"* (F3). Some participants however described using self-management strategies such as stretching; *"it will remind me to do the stretching that I know I should be doing anyway"* (M1). Additionally, some runners may look to external sources such as YouTube for appropriate stretches/exercises; *"I find myself YouTubing quite a lot. If I have a pain in my calf, I just do a five minute exercise that I find on YouTube and it will be gone that day"* (M7), or casually chat to their running friends about the complaint: *"but only by the by, if it came up in conversation I might mention it"* (F10).

**Niggle.** 'Niggle' was a term used by all participants in all focus groups with different descriptions presented. In its mildest form, some runners physically described a 'niggle' as being more *"aware"* (M13) or *"conscious"* (F8) of a certain body part, describing it as *"not* [being] *the same as the other side* [of the body]*"* (M1), or suggested it is *"background noise"* (F10). Other runners described a 'niggle' as something slightly more severe, with phrases such as *"background pain"* (F15), and *"a pain that shouldn't be there"* (M4). The majority of participants described a 'niggle' as something *"repeated"* (F8), *"persistent"* (F3) and *"something that lingers"* (F11): *"discomfort can come and go, whereas a niggle is sort of always there"* (F10). In relation to the outcome, the majority of participants continue full training with a 'niggle', with the perception that runners are *"quite high functioning with a niggle"* (F17). Many runners described a 'niggle' as *"not bad enough"* (F7) to stop training, and suggested that they can *"cope with a niggle"* (F11), *"ignore it a lot of the time"* (F13), or they can *"run through it"* (F18). Some also described how they feel pressure to continue training with a 'niggle', whether it is due to running with a group: *"if I am with a group I am like 'oh I need to keep going', but I'd probably be making it worse"* (F11), or whether it's a personal desire to continue:

> *"you are chasing the high, chasing the endorphins. When you just get that niggle. . . you really just keep pushing yourself. . . And then because it is not an external thing, you can't see it, you can just kind of ignore it"*

> (F13)

However, this level of injury can begin to affect running for some runners, causing them to *"start reducing* [their] *mileage because the niggle has hit"* (F17), or to complete a more vigorous warm up because they feel *"I should warm up properly if I have a niggle"* (F17). There was no effect to a runner's daily life, but as a psychological description, some suggested they are *"cautious"* (F5) of a 'niggle' and that it can affect their motivation to train; *"it makes the run harder to complete mentally, rather than it stops you running"* (F6).

Some runners don't practice any management strategies because they suggested that 'niggles' can resolve on their own, they *"get away with a lot of niggles"* (M4) or they can *"ignore it"* (M11). However, some runners use self-management strategies and alter their training, describing how *"niggles encourage me to stretch"* (F15). Some may also *"take a day or two off"* (F15) or reduce their training load, *"not run as far"* (F2), or *"slow down"* (F5). Some may also use additional therapeutic modalities such as foam rolling; *"You know that roller that is over in the corner of the front room that you occasionally use, that is when you use it"* (M3). Some participants described that they would also turn towards external management strategies, such as online resources or asking their running friends for advice or support; *"it might come from an external source. . . somebody says, 'maybe you should have that looked at'. . . if you complained about it enough to somebody else, rather than deciding yourself"* (F9).

As well as describing a 'niggle' in terms of its physical and psychological descriptions, its outcome, and management, many participants reported that 'niggles' are *"inevitably a*

*fact of running"* (F6), that *"runners constantly have niggles"* (F16) and they live in *"Niggle City"* (F17). Many participants did not perceive a 'niggle' to be an 'injury', but described that it can contribute to the development of an 'injury'; F16—*"it could turn into an injury, but I wouldn't consider it an injury"*. Many participants reported that they perceive 'niggles' to be an *"early warning sign"* (M11), a *"potential injury"* (M13), or the *"root to an injury"* (F17).

**Twinge.**    A 'twinge' was described as a *"darting pain"* (M7) or an *"intense, quick, sharp pain"* (F6), that is intense enough to cause a runner to *"stop and walk"* (F3) mid-session, or to stop their session completely; *"I might stop if I had a darting pain"* (F4). However, a twinge was described as something temporary, short-lived, and typically a once-off; *"by the time it happens, it is gone"* (F3). Runners described that it would not be felt during the next session, and they would *"forget about it"* (M7). There was no description of an effect on a runner's daily life, but participants associated a 'twinge' with *"caution"* (F7).

In relation to management, all runners use some sort of self-management strategy by this level; *"if I had a darting pain I would do something. I wouldn't just keep on running with that one"* (F4). Some runners described that *"stretching would come in here"* (M11), they would *"stop and walk"* (F3), or do *"slower runs"* (F7) at the onset of a 'twinge'. Some participants also suggested they would continue to consult friends, asking *"what do you think? What would you do?"* (M9), and online resources *"you would definitely Google it"* (M9).

**Persisting niggle.**    A 'persisting niggle' was described as a progression of a 'niggle', however, it differs because it was described as more severe and more persistent. Progressing from milder terms (such as tightness or tiredness) used to describe a 'niggle', a 'persisting niggle' was associated with a description of pain, with *"low-medium"* (F2) pain and *"mild pain"* (F12) being suggested. It was also described as *"persistent"* (M5) and as occurring on *"consistent occasions, consistent runs"* (F4).

> *"Having the same niggle a few runs in a row, where you know it is not a niggle anymore. If it happens again the next week, and the next week, it doesn't become a niggle anymore, it becomes a problem where you know it is not going to go away"*
>
> (M7)

In relation to the outcome, some runners will continue to train fully at this level because *"it is tolerable to keep running"* (F6) and they want to *"take a chance"* (M7) and hope that it will not progress to a further level. However, the majority of runners will change their training at this level by *"decreasing [their] load"* and *"hopefully [going] back to no injury"* (F9). Some runners may also take additional rest days; *"I might stop and take a little bit of a rest, but be back at it. The pain wouldn't need to go. It would just need to be a bit better and then I would go again"* (F4). There was still no description of this level affecting a runner's daily life, however, in relation to the psychological description, this level inspired the first mention of associated concern and *"anxiety"* (F7).

With regard to management, self-management strategies progress from previous levels with further alterations to training being made. Some runners will *"decrease [their] load"* (F9) because they *"can't go full tilt"* (F2), while others will *"take a break for a few days"* (F3). As external strategies, there is a continuation of consulting friends and getting *"peer advice"* (F6), as well as using the internet: *"using Google because it's the go-to"* (M9). This level is associated with the first mention of obtaining medical professional attention from a Physiotherapist/ Athletic Therapist. Some participants described that they would be *"looking for physio support"* (F2) because they *"probably should go and get it looked at"* (F2).

**Non-responding niggle.**    A 'non-responding niggle' was described as the point at which all attempts to manage complaints thus far have failed, and runners are at a *"crossroads"* (M4) because their 'injury' is *"not responding"* (M4). Participants described how it is more evident that they need to make a decision at this level of whether they continue running (with altered training), stop running (for an unknown period of time), or seek external medical attention.

Physically, this level was described as causing *"more intense"* (F15) pain that is *"getting worse"* (F16) and is increasingly persistent to the point of being *"constant"* (F16). It was described as pain that *"doesn't stop"* (F4) despite management attempts being made. In relation to the outcome, some runners will continue training, although with a reduced training load, because they are "*not prepared to leave*" (M3). Furthermore, some runners normalised running with this level of injury and suggested that *"everybody runs with an injury. . . I have never met anybody who didn't run with an injury"* (M9). However, others suggested that they *"shouldn't be running"* (F3) at this level, and that they will stop training in the short-term: *"middle of the road to me is where we are going to rest for a week"* (M9). This level was the first mention of a negative effect on a runner's daily life, with description of *"pain filter*[ing] *through the rest of your day"* (F9) and pain being present *"as you're walking around"* (F18). Participants suggested that they will begin to make conscious decisions and efforts to protect this level of injury; "*I might choose to bring the dog to a field and throw a stick and let him run, rather than me having to walk the 4km with him"* (F15). With regard to a psychological description, runners would become increasingly anxious at this level, thinking *"do I need to worry here?"* (F16).

In relation to management, some runners continue with self-management strategies by choosing to take a short-term break from training; *"I would rest myself. Ease up for a week and see how it felt"* (M9), while others will seek external advice because these are *"the injuries where we run out of ideas of how to treat it* [ourselves]*"* (M3) and they are at *"the point where you go to a physio"* (F15). Some participants also described how they would still consult their running friends or *"someone who has a lot of running knowledge"* (M7) for advice or support.

**Short-term injury.**    A 'short-term injury', was described as causing *"severe discomfort"* (M6), *"dull pain"* (F12), and *"really sharp pain"* (M10). It was described as causing runners *"constant pain"* (F12) if they continue to run, and getting progressively worse; *"I'll keep going and make it worse until I have to stop"* (M13). Additionally, some runners perceived that this level would cause a physical sign or a *"visible effect"* (F6) of an injury, such as limping or swelling. In relation to the outcome, this level will cause a *"short-term stoppage"* (F6) to running (i.e., days/weeks) for the majority of runners, and was referred to as a *"stopping injury"* (M13) where they cannot continue to train. It will also continue to affect a runners' daily life, with *"pain outside of running, pain in work"* (F12) and having an effect on decisions such as *"taking the car instead of walking somewhere"* (F7). Runners are also becoming increasingly anxious at this level and describe that an injury which *"stops you running. . . really messes with your head"* (M7).

As well as being described as a progression along the Running Injury Continuum, some participants described this level of injury as one which has an acute onset (e.g., muscle strain or joint sprain), causing a short-term stoppage to running (i.e., for a number of weeks); *"a sudden injury where you just have to come to a standstill"* (F5). At this level, external management strategies take over as the primary method of management, with participants describing that they *"need some sort of intervention"* (M6). Participants suggested that Physiotherapists/Athletic Therapists are their primary sources of medical intervention, and they feel they "*have to go to physio now because it is not going away"* (M11). Few runners will continue to self-manage by taking a short-term rest (i.e., weeks) from training, and described this level as the *"point you need to realise you have to rest, cut back"* (F12).

**Long-term injury.** With 'long-term injury', there was less emphasis placed on the physical description, although some runners still described *"extreme discomfort"* (M6) and *"very high pain"* (M6), with a greater focus on the outcome. At this level, all runners have stopped running in the long-term (i.e., months or longer) because they are unable to run and describe being *"out of action for a few months"* (M3). Similar to the previous level, runners' daily lives are affected and conscious decisions are made to offload the injured area. As a psychological description, runners are increasingly anxious and becoming frustrated; *"the injury is about the frustration of not being able to do what you want to do"* (M9).

By this level, all participants are using external management strategies, with runners turning towards interventions from medical specialists or *"the correct doctor"* (M6). Some will continue to consult an Physiotherapist/ Athletic Therapist to dictate the most appropriate course of action, however, runners described that this level requires specialist intervention;

> *"When I have gone to the physio four times, five times, and the physio says 'look, what I have done should have helped it, it is not helping it, so there is obviously something else wrong here, so my advice is you need to get referred to a consultant'"*

(M11).

**Career-ending injury.** While no participant reported experiencing a 'career-ending injury', this was the perceived as the most severe injury a runner could sustain, and a suitable end-point for the Running Injury Continuum. With this level, there is less attention on the physical description, although severe pain and *"the worst possible pain you could imagine"* (F6) was suggested, and more significance to the outcome. This is a level which *"stops you running, forever"* (F3), is a *"permanent"* (M10) injury and a *"show stopper"* (M3). Some suggested that this level would significantly affect their daily life where *"you can't do ordinary stuff, even in your household duties"* (M5), with associated feelings of frustration and depression regarding their injury.

Some runners described using external management strategies as final attempts to manage this level of injury, with *"need*[ing] *surgery"* (F4) being suggested as a potential strategy. However, as the worst possible injury suggested by runners, it was described that they would never run again.

## Individual perception

It is important to note that many participants suggested that runners' description of injury and their management of the process of injury development is *"not the same for everyone"* (M8) and *"there are a whole host of external factors that inform your perception of it"* (F10). Such factors include their running habits and history, individual factors, and injury-related factors (Table 3).

With regard to running habits and history, some participants suggested that description and management of injury is *"based on your experience"* (M9) and *"the length of time people are running, or the experience* [they] *have with injury"* (F17). Participants suggested that more experienced runners will have a better understanding of the levels of injury, and manage these levels more appropriately: *"newbie runners. . . they don't know what a niggle is"* (M2). Additionally, participants described how a runner's *"motivational factors"* (F9) influence their management of the process of injury development: *"it boils down to what your objectives are"* (M10). Runners suggested that those who are more competitive or those training for a specific

**Table 3. 'Order of themes' document: Factors that influence runners' description and management of the injury development process.**

| Core categories | Themes | Sub-themes |
|---|---|---|
| Running habits & history | Running experience | |
| | Motivations | Competitiveness |
| | | Chasing high |
| | | Goals |
| | | Stick to a plan |
| | Other sport participation | |
| | Run setting | Group setting |
| | | Race |
| | | Individual |
| | | Training session |
| | Event coming up | |
| | Knowledge | |
| Individual | Individual perception | |
| | Daily life | Children |
| | | Mood |
| | | Menstrual cycle |
| | | Fatigue |
| | Age | |
| | Sex | |
| Injury | Previous injury | |
| | Type of injury | |

goal will continue training with a *lower level injury*, rather than reducing their load because *"sometimes the benefits just outweigh the risks"* (F12).

In relation to individual factors, participants suggested that description and management of injury *"depends on the person"* (F15), and can vary from runner to runner: *"some of your definitions of niggles would not be mine"* (M1). Some participants also discussed how a runner's daily life can influence their description and management of injury, with factors such as *"state of mind"* (F3) on a particular day or their menstrual cycle making someone feel *"sluggish"* (F6) influencing their description and management of injury. Additionally, some female participants suggested that having children to care for will influence their management of injury because they *"can't afford to be laid up in a bed"* (F2):

> *"the person who doesn't have children, or can have all that time to rest before and after the run, they might be more likely to do the run* [while having a lower level injury]... *If you had to come home and you go, 'right, if I go for a run this morning I will not be able to do the nursery football with the children after school because I will be in pain'"*

(F9)

Finally, participants also suggested that injury-related factors, such as previous injuries and *"how impacted you have been by injury in the past"* (F11) will influence how a runner describes and manages all levels of injury. Participants described that they would *"intervene earlier if it is something* [they] *have had before... and go quicker through the* [management] *steps"* (F8), compared to an injury they have never had.

## Discussion

This study provides a qualitative insight into how recreational runners describe injury and manage the process of injury development. By capturing the lived experiences of runners, the authors present a comprehensive representation of RRIs, highlighting their progressive, over-use nature. The current study used an IPA to explore this topic, and the authors cannot over-state the richness and depth of data that was captured, primarily facilitated by the social interaction between participants during focus groups.

### The Running Injury Continuum

The Running Injury Continuum (Fig 1) reflects runners' descriptions of RRIs from injury-free to career-ending injury, and is made up of nine levels of injury, each increasing in injury severity. The nine levels are categorised into *lower* and *higher level injuries*. *Lower level injuries* span between 'discomfort' to 'non-responding niggle', while *higher level injuries*, which are most associated with the RRI consensus definition [16], span between 'short-term injury' to 'career-ending injury'. The consensus definition defines a RRI as: "running-related (training or competition) musculoskeletal pain in the lower limbs that causes a restriction on or stoppage of running (distance, speed, duration, or training) for at least 7 days or 3 consecutive scheduled sessions, or that requires the runner to consult a physician or other healthcare professional" [16, p.377]. *Higher level injuries* overlap with this definition by virtue of the commonalities between our participants' description and the criteria used in the consensus definition. Firstly, a description of pain is associated with *higher level injuries* and is required to define injury within the consensus definition [16]. Although pain is mentioned in earlier levels of the Running Injury Continuum, it is described in milder forms and becomes significantly more prominent at these *higher level injuries*. Secondly, within *higher level injuries*, runners will stop running, at least in the short-term (e.g., a week), a criterion and time-frame strongly associated with the consensus definition [16]. While earlier *lower levels of injury* were described as causing restrictions to running (such as reducing load), they were not associated with this length of a time-frame, distinguishing them from the consensus definition. Finally, *higher level injuries* result in runners requiring medical attention, an evident criterion in the consensus definition [16]. It can be argued that descriptions of the 'non-responding niggle' (medium pain causing alterations to training, short-term rest, or HCP intervention) are consistent with elements of the consensus definition; however, this level is not included as a *higher level injury* for two reasons. Firstly, the opinions of participants varied across the categories of descriptors (e.g., some described continued running, whereas others described a stoppage to running), with more agreement being evident within *higher level injuries*. Secondly, there was a strong description of this level being a 'crossroads', more so as a transition level from *lower* to *higher level injuries*. Runners described each level of injury using four categories of descriptors: physical description, outcome (the effect on running and on daily life), psychological description, and management (self-management, and external management strategies). This is a bi-directional continuum, on which runners can progress or regress, either increasing or decreasing in injury severity depending on their management of each level. During injury development (or recovery), runners do not have to progress through the immediate succeeding (or preceding) level of injury (e.g., runners can progress from a 'niggle' straight to a 'non-responding niggle').

Both end-levels of the continuum (i.e., 'running smooth' and 'career-ending injury') were clearly described by participants, with a high level of agreement achieved amongst all participants. However, there was variance in opinion regarding the seven levels in-between, with some overlapping descriptions across adjacent levels. This is captured in the term 'continuum' which is "a continuous sequence in which adjacent elements are not always perceptibly

different from each other, but the extremes are quite distinct" [49]. It also reflects runners' perception of the progressive and regressive nature of RRIs. While the categories of descriptors used to differentiate levels of injury were sometimes not unique (e.g., caution was used to psychologically describe both 'niggle' and 'twinge'), it was possible to differentiate between levels of injury by comparing across *all* categories of the descriptors used (e.g., 'niggle' was described as a repeated low pain, whereas 'twinge' was described as a temporary darting pain). The term 'niggle' was the most commonly used term to suggest a *lower level injury*, used by every participant in every focus group. From our findings, a 'niggle' can be defined as: *'a repeated physical sensation (discomfort or low pain) with which a runner can continue to run'*. However, the level 'discomfort' is the first level of a complaint along the Running Injury Continuum and initiates the progression of *lower level injuries*. From this complaint of temporary discomfort or tightness, the Running Injury Continuum advances into three distinct levels of progressive and worsening 'niggles': the 'niggle', the 'persisting niggle', and the 'non-responding niggle'. These three entities describe levels of injury which become increasingly more severe in terms of their physical description, their outcome (effect on running and daily life), their psychological description, and the management strategies required. However, amongst these three levels of niggle, there is a further *lower level injury* that does not follow this progressive nature; the 'twinge'. A 'twinge' can be defined as: *'an acute onset of pain resulting in an immediate outcome (either a reduction in training load within a session, or the stoppage of a training session), but which does not persist to the next session'*.

The Running Injury Continuum supports previously published representations of overuse injuries. A comparison can be made between our participants' description of escalating levels of injury severity and the Well-Being Continuum [17], which describes the escalating levels of biological and physiological tissue damage associated with overuse injury development. Our findings also support the Injury Pathway which represents runners' own views on the process of RRI development [19]. While other papers have implicitly referred to an 'early phase of injury' during the injury development process, using phraseology such as "early phase" [15], "early symptoms" [21] and "injury category" [22], to the best of our knowledge, only one study has explicitly identified and named a *lower level injury* during this process: the 'complaint' stage of the Injury Pathway [19]. However, our study explores this to a greater extent and appears to be the first to explicitly provide sub-categories within this phase. Additionally, rather than concluding the process of injury development at a single point termed "injury" (as with the Injury Pathway [19]), we have identified further sub-categories within this later phase (i.e., *higher level injuries*) which map with the consensus definition [16]. Furthermore, we provide a rich and in-depth account of runners' description and management of the process of injury development, in both the early and late phases.

The OSTRC Overuse Injury Questionnaire (OSTRC-O) is a widely cited tool for surveilling overuse injuries in sport research [18]. It acknowledges *lower level injuries* in that it recognises the importance of non-time loss injuries, and does so through monitoring both the characteristics of pain (physical descriptor) and effect on running (outcome) [15]. However, our findings build upon the OSTRC-O, indicating the importance of capturing additional categories of descriptors, including the psychological response to injury and the management strategies used, both of which can impact injury development [19, 23, 50]. Additionally, a limitation to the OSTRC-O is that it does not capture injuries from a 'traditional' definition point (such as that defined by the consensus definition [16], injuries which are clearly described and experienced by runners. It has been suggested that the development of a single tool capable of monitoring the continuous development process of overuse injuries (as done by the OSTRC-O) as well as registering injuries from a more 'traditional' point (i.e., time-loss) is warranted and could greatly assist injury surveillance research [51]. We suggest that the Running Injury

Continuum may provide a basis for such a tool. Additionally, in studies investigating risk factors for injury, it may be beneficial for researchers to determine the specific level of injury (e.g. niggle) experienced by athletes, as these *lower level injuries* have the potential to not only interact with other possible risk factors for injury (increasing injury risk), but also to potentially act as risk factors for injury themselves. By using regular surveillance and capturing this level of detail, researchers could be provided with significant insight into: the risk factors for RRIs, the development process of RRIs, and understanding how the consequences of injury change during this process.

Our findings highlight the importance of the psychological response to injury experienced by runners. It is well accepted that athletes may experience psychological distress in response to injury [52, 53]; however, due to the insidious nature and longevity associated with RRIs, runners often experience significant and prolonged psychological distress during the injury development process [54, 55]. Specifically, our findings suggest that runners experience a progressive psychological response to injury that increases in severity as the Running Injury Continuum progresses. Our findings support previous research which highlights runners' experiences of psychosocial distress in response to overuse injuries [23, 55, 56], with reports of frustration, fear, general psychosocial distress, and social influences experienced by runners during the injury process [55]. However, in contrast to previous research which identified that these responses occur from injury onset (defined as the point where runners perceive themselves to be injured, or pain is affecting their running) [56], our findings indicate a psychological response that occurs from an earlier phase during the injury development process. We identified the first description of a negative psychological response to injury at the level of a 'discomfort', where runners describe 'mental fatigue'. This level identifies the start of a pathway of worry, concern and anxiety experienced by runners during the injury development process. The capture of runners' description of anxiety during these *lower level injuries* is a novel finding as, to the best of the authors' knowledge, previous research has not reported such an extreme psychological response at such an early phase of injury development. It is crucial that clinicians, coaches, runners and other personnel involved with runners' well-being are aware of this finding and understand the level of psychological distress experienced by runners, especially during the early phases of injury development.

### Management

Our results support previous research which identified runners' desire for autonomy in the dealing with *lower level injuries* by predominantly using self-management strategies, and concur with the finding that once runners lose this autonomy and require external professional assistance, they perceive themselves as 'injured' [19]. Typical self-management strategies involve reductions in training load and using therapies such as ice, stretching or general rehabilitation exercises to prevent or slow the progression of their *lower level injury*, while maintaining some level of training. Our findings also agree with previous research which reports runners' reliance on non-evidence based sources of information (such as web-based or peer advice and previous personal experience) to inform self-management of their *lower level injuries* [19, 23, 55, 57], as well as their reluctance to attend healthcare professionals (HCPs), despite experiencing physical and psychosocial distress [55]. Participants suggested several reasons for not attending a HCP. Firstly, *lower level injuries* are not severe enough to warrant HCP input. Secondly, some runners described wishful thinking regarding their *lower level injury*, hoping that it will resolve on its own without the need for HCP intervention. Wishful thinking is a bias pervading the management of persistent musculoskeletal pain, where decisions and beliefs regarding an injury are based on what is pleasing to imagine, rather than

based on evidence, rationality or reality [58]. Finally, runners suggested that their previous experience with injury removes the need to attend a HCP. This finding is similar to previous research in which runners have reported that those with more running experience are better able to self-manage RRIs [19, 23]. It has also been suggested that novice runners are more at risk for RRIs [59], while coaching or education (i.e., increasing runners' understanding of the injury development process) is theorised to enhance injury prevention [60].

The decision to seek HCP advice most often came at the level of 'short-term injury', and from this point on, as the injury becomes more severe or impactful, HCP advice becomes more specialised. Runners described several reasons for seeking HCP advice. Firstly, when their attempts to self-manage injury had failed, or their injury had become too severe where they can no longer self-manage, similar to previous running-based research [23]. Secondly, they would attend a HCP because they are seeking validation of their injury, typically in one of two ways. Runners are either seeking confirmation that they are actually injured and the injury is not in their head (as often, there are no physical signs of injury and they may be able to continue training); or they are looking for reassurance that their injury is not as serious as they may be concerned about, and they are seeking guidance on continuing their training.

## Implications

Our findings have several implications for HCPs, coaches and researchers.

**Education.**   Support for the relationship between education and injury prevention (IP) has been described in terms of the translation of knowledge to enhance the adoption of IP interventions [61], and enhancing the recovery process to prevent injuries [62]. Research specifically examining this relationship involving runners seems to be limited; however, an online IP intervention consisting of educational videos informing participants about the aetiology and mechanisms of injury, combined with evidence-based IP advice was shown to have a positive effect on knowledge, attitude, intention, and behaviour [63]. Additionally, runners' perception of injury risk and their attitudes towards the importance of executing IP measures were positively affected by the intervention, which included these educational messages [63]. Furthermore, another study examining the effectiveness of an online IP intervention found no significant effect of their intervention on actual preventative behaviour [60]. One suggestion for this was a difference in the content of the IP interventions, such as the educational videos which were included in Adriaensens and colleagues' [63] study [60]. These findings highlight, that with enhanced knowledge and education regarding injury risk and management, runners are more likely to adopt IP practices.

From our findings, it is clear that the education of runners regarding evidence-based information on managing and preventing RRIs is required. Firstly, HCPs should be aware that runners typically do not attend an HCP with a *lower level injury* because they believe they can be primarily self-managed. While this is a positive finding in the sense that runners feel empowered to self-manage their own *lower level injuries*, it is also clear that HCPs need to educate their patients on appropriate self-management strategies, by directing them towards trustworthy sources of information, ensuring runners are using evidence-based recommendations to prevent and rehabilitate injury. Similarly, coaches, running clubs, and governing bodies need to educate runners, encouraging them to use evidence-based practices in the management of their injuries (e.g., Athletic Therapists/ Physiotherapists, evidence-based sources of information). Secondly, there is a need for enhanced dissemination of evidence-based information to runners. Researchers need to ensure their findings are freely accessible to runners, disseminating findings in user-friendly formats (e.g., infographics, podcasts, blog posts) using plain language, ensuring runners understand key information.

**Appropriate monitoring of RRIs.** Our findings highlight the importance of a wider scope of monitoring RRIs, not just across all levels of injury, but across all categories of descriptors. Firstly, HCPs should consider the potential importance of *lower level injuries* acting as risk factors for *higher level injuries* (as discussed above), and appreciate that a runner presenting with a *higher level injury* may have had a preceding *lower level injury*, in order to better manage the whole continuum of injury. Secondly, when designing injury management strategies, HCPs should understand that runners will likely have made attempts to self-manage their injury prior to presenting to them, and these attempts should be taken into consideration. Finally, with the emphasis placed by participants in the current study on the psychological description of each level of injury, it is crucial that HCP's include biopsychosocial assessments when dealing with recreational runners, ensuring this psychological response is captured, monitored and incorporated into management strategies. HCPs should also educate their patients regarding the psychological aspect of the process of injury development, and ensure patients understand that it is normal to experience these responses (i.e. to experience anxiety or concern, especially with a *lower level injury*) [55]. Similarly, in helping athletes manage injuries, coaches need to be aware of the importance of not only monitoring all levels of injury and supporting athletes with appropriate management strategies, but also on monitoring athletes across all categories of descriptors.

Future research should also broaden its scope of investigating RRIs (and overuse injuries) to ensure that all categories of descriptors are captured in order to better understand the wider impact of an injury. In particular, this will allow examination of whether *lower level injuries* are risk factors for injury, and/or how they interact with other potential risk factors.

## Strengths and limitations

A representative sample was included in the current study, gathering the perceptions of runners of various ages and running backgrounds. We included a larger sample size of runners compared to previous research [19, 23], to ensure the broad scope of the study was explored in detail, and to enhance the reliability of our findings. Richness and depth of data was facilitated through focus groups guided by IPA [31]. Furthermore, as IPA considers participants' lived experiences as well as how they reflect on these experiences, it enhanced our interpretation of their lived experiences of the injury development process, in the forum of the Running Injury Continuum [64]. Additionally, we employed a method of constant comparative analysis throughout the data collection and analysis phases, enhancing the methodological rigour. Multiple coders with different research and lifestyle backgrounds were involved, reducing the potential of researcher bias. Furthermore, several methods of trustworthiness were executed to ensure appropriate interpretation of findings and enhance the credibility of results.

The study's findings should also be considered in light of some limitations. Our sample consisted of only Irish runners, therefore these findings may not be representative of the global population of runners. In particular, specific terminology used (e.g. niggle) or methods of management employed (e.g., access to clinicians) may not be consistent across other nationalities or socio-demographics. Furthermore, as we only recruited recreational runners, findings may differ with elite or novice runners. As the aim of this study was to explore the lived experience of runners, it was necessary for participants to have previous experience with *lower level injuries*; however, it is possible that different findings would be reported from runners with less/no experience of *lower level injuries*. Given the adopted terminology of 'niggle', 'persisting niggle' and 'non-responding niggle', a 'twinge' can be viewed as a distinct level from these because of its associated sharp pain, its immediate effect on running, and its presence for only one session. The location of 'twinge' between 'niggle' and 'persisting niggle' or 'persisting

niggle' and 'non-responding niggle' was not consistent across all focus groups; however, its placement on the Running Injury Continuum was arrived at because all participants described it as something more severe than a 'niggle', while the majority described it as less severe than a 'persisting niggle'.

## Conclusion

Through capturing the lived experiences of recreational runners, we present the Running Injury Continuum as a representation of the development process of RRIs. Expanding on previous research [19–23], nine distinct levels of injury were identified in the current study, with each level being described across four categories of descriptors: physical description, outcome, psychological description, and management. For research purposes, the Running Injury Continuum is a tool that can be used in both injury surveillance research and research investigating risk factors for RRIs.

Our findings clearly highlight the importance of education and accessibility of evidence-based information. HCPs need to educate their patients on appropriate self-management strategies for RRIs, while researchers should ensure recreational runners have access to evidence-based information, and can utilize this information in their running practices. HCPs, coaches and researchers should broaden their scope of monitoring RRIs to ensure that all levels of injury and categories of descriptors are captured, in order to better understand the wider impact of RRIs, to more appropriately manage RRIs, and potentially enhance injury prevention.

## Supporting information

**S1 Table. Focus group schedule, introduction and aims.**
(DOCX)

**S2 Table. Standards for Reporting Qualitative Research (SRQR) checklist [38].**
(DOCX)

**S1 Fig. Data analysis and the Running Injury Continuum development process.** AL: Aisling Lacey, SOK: Sinéad O'Keeffe, EW: Enda Whyte, KM: Kieran Moran, SOC: Siobhán O'Connor, AB: Aoife Burke.
(DOCX)

## Acknowledgments

The authors would like to thank the focus group participants for their contributions.

## Author Contributions

**Conceptualization:** Aisling Lacey, Enda Whyte, Kieran Moran.

**Data curation:** Aisling Lacey, Sinéad O'Keeffe.

**Formal analysis:** Aisling Lacey, Sinéad O'Keeffe.

**Funding acquisition:** Kieran Moran.

**Investigation:** Aisling Lacey, Sinéad O'Keeffe.

**Methodology:** Aisling Lacey, Sinéad O'Keeffe.

**Project administration:** Aisling Lacey.

**Supervision:** Enda Whyte, Kieran Moran.

**Writing – original draft:** Aisling Lacey, Enda Whyte, Kieran Moran.

**Writing – review & editing:** Aisling Lacey, Enda Whyte, Sinéad O'Keeffe, Siobhán O'Connor, Aoife Burke, Kieran Moran.

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
