## [Decision Letter · Decision Letter 0]

21 Aug 2023

PONE-D-23-10426

The Running Injury Continuum: A qualitative examination of recreational runners’ description and management of injury

PLOS ONE

Dear Dr.Lacey

Thank you for submitting your manuscript to PLOS ONE. After careful consideration, we feel that it has merit but does not fully meet PLOS ONE’s publication criteria as it currently stands. Therefore, we invite you to submit a revised version of the manuscript that addresses the points raised during the review process.

Please see comments below.

We look forward to receiving your revised manuscript.

Kind regards,

Daniel Ramskov, Ph.D

Academic Editor

PLOS ONE

Journal Requirements:

Additional Editor Comments:

Dear authors

Thank you for the effort put into this study. I really enjoyed reading the article and read the results part with great interest. I believe that the work put forward here is relevant and important.

It has been a challenge securing external peer review, which is why only a single external reviewer has commented on the manuscript. Since the journal policy is at least two external peer-reviews, I have also provided extensive reviews and comments to accommodate for this.

Please excuse the sometimes straightforward communication, no offense is meant.

Introduction

Regarding references 1-3, consider changing these references or including this PMID: 27939265.

Regarding references 4,5, and 6. Consider the terminology and use of the references. Reference 4 is specific to differences in risk between genders, not just high rates of injury. Reference 5 reports both incidence and prevalence and includes region-specific injury proportions. Consider the relevance of reference 6 since it´s an original study reporting a single identified injury proportion from a cohort of runners.

Furthermore, reference 6 is included in reference 5, and reference 5 is a literature study reporting injury proportion across a range of studies. Therefore, it may be a more accurate estimate of the frequency of injuries. It could also be relevant to include the following reference PMID:25951917 maybe. In that way, you could very nicely describe the nature of RRI regarding frequency, gender differences, performance level differences, and differences between anatomical regions.

Regarding reference 7, I´m not able to see how this reference supports the notion that most RRI are overuse injuries. I agree that the vast majority of injuries, indeed, are overuse injuries, I just think the reference are inappropriate.

Regarding Figure 1 and the introduction part lines 74-89. The thoughts presented here are very theoretical and somehow disrupt the flow of an otherwise coherent introduction. In the discussion, this subject is also included. Consider removing it from the introduction.

Methods

Several changes need to be made in the design section. First, the applied methodology should be described in the beginning. Secondly, the method of data acquisition could be described.

Much more effort needs to be directed toward describing how the data acquisition method fits the methodological framework. Several of the referenced papers in this section elaborates upon exactly this issue, but the description seems superficial and standardized.

The choice of focus groups is not a problem. However, no consideration is given to how the interaction between the group members is handled. This is especially relevant in terms of analysis, and from what I can understand, video recordings of the interviews exist. Both reference 35 and 37 provide insight and discussion surrounding this issue.

Regarding reference 36, I´m not able to see how this reference supports the notion that focus groups are unsuitable for phenomenological research.

Please be explicit and precise about the choice of methodology instead of only elaborating slightly upon interpretative phenomenology.

Lines 117-119 should be moved to study procedures.

Lines 119-121, starting with ”The schedule……..” should be moved to pilot study or removed.

Lines 132-133 describe that all participants reported a lower level injury. However, in the result section lines 253-256, it is described that not all participants have experienced a RRI. I assume it is because a distinction between a RRI and a lower level injury is made. But, chronologically, the reader is not familiar with a lower level injury based on the developed running injury continuum. Therefore, this may be more appropriate elsewhere, maybe in the result section….

Please provide further information on the participants in the individual focus groups. And elaborate on if the 2 groups used in the pilot study are included in the total sample of 7 groups.

Line 191-193 mentions the development and expansion of themes. This warrants further elaboration. Consider including supplementary material C in the article instead of Figure 2. Because supplementary material C provides valuable insight into the data analysis process, and Figure 2 is very superficial. Further, supplementary material C is mentioned several times in the manuscript, which further highlights its relevance.

Results

Table 1, is very hard to read. Please consider rearranging the table. I would suggest having the headlines as columns and the categories as rows.

Consider changing athletic therapist to physiotherapist in the figure, since all included narratives in the manuscript mentions physiotherapists.

Discussion

More precision is needed in discussing how and where the RRI consensus definition overlaps with the presented running injury continuum. Especially considering the perspective put forward in relation to both clinical practice and future research.

When discussing the education of runners from a preventive standpoint, please include and discuss some of the available knowledge on the effect of educational interventions to reduce the risk of RRI, relative to the running injury continuum.

The validity of the running injury continuum needs thorough discussion. The sample size is highlighted, but data saturation is not discussed. Why is that? The following reference may provide useful information for the discussion PMID: 25951917

A discussion/sharing of experiences about how the interpretative phenomenological approach and using focus groups as a method of collecting data is warranted.

Reviewers' comments:

Reviewer's Responses to Questions

**Comments to the Author**

1. Is the manuscript technically sound, and do the data support the conclusions?

Reviewer #1: Yes

2. Has the statistical analysis been performed appropriately and rigorously? 

Reviewer #1: Yes

3. Have the authors made all data underlying the findings in their manuscript fully available?

Reviewer #1: Yes

4. Is the manuscript presented in an intelligible fashion and written in standard English?

Reviewer #1: Yes

5. Review Comments to the Author

Reviewer #1: I thank the editor for the opportunity to review this manuscript. This study explored the perspectives of runners regarding running-related injury and their management strategies when managing injury. This qualitative study is well-justified and contributes to filling a gap in running-related injury research where further qualitative research is needed. The authors should be commended for undertaking this work with quite a large sample size, which can involve substantial work in the context of the thematic analysis method. Additionally, I am pleased to see that this study explored perspectives of both male and female sexed runners.

A novel aspect of this study is the description of a ‘running injury continuum’ which is an astute observation and interpretation of the available data. In particular, this continuum represents well both the experiential aspects of RRI (of different levels) but also links these individual (but overlapping) levels and the kinds of health-seeking behaviours which runners may display.

I have several minor comments which are provided in a genuine attempt to improve aspects of this manuscript:

First, the authors have done great work describing a new injury continuum and a new theoretical model for injury development. However, the continuum depicted in figure 1 seems a bit simplistic, given the multifactorial aetiology of RRI. I wonder if this model could be revised to include more preceding factors other than simply a change in running biomechanics (which have a debatable association with RRI aetiology).

Introduction line 65-68: “and the identification of a ‘complaint’ level in the process of injury

development (prior to time-loss, training restriction or seeking medical attention) (22).” This sentence feels incomplete or may require grammatical revision.

Introduction line 72: perhaps ‘gradual and multifactorial’ would be better phrased as ‘multifactorial and progressive’, at the authors’ discretion.

Methods/design line 121-122: is there an ethics approval number which should be listed?

Line 122 “were adhered to”

Participants, line 132-133: can the authors describe how they confirmed all participants had experienced a ‘lower level injury’?

I apologise if I have missed this somewhere throughout the manuscript, but could the authors clarify how the participant demographic information was captured? Was this through a paper-based survey? Are all survey data presented in the manuscript? Please also include the questioning methods e.g. Likert scales

Whilst reading the results, I quickly identified that there is overlap between individual levels of injury with respect to self-management, peer support, and health-seeking behaviours, and this is very well addressed by the authors in the discussion.

Completely discretionary comment, but I wonder if the terms ‘short term injury’ and ‘longer term injury’ would be more easily adopted than ‘injury: short-term effect/long-term effect’?

Career ending injury – had any of the runners experienced a ‘career ending injury’ which caused permanent cessation of running?

Line 564 “not individually unique” suggest alternate phrasing as ‘individual’ and ‘unique’ sound quite synonymous

6. PLOS authors have the option to publish the peer review history of their article (what does this mean?). If published, this will include your full peer review and any attached files.

Reviewer #1: **Yes: **Dr Benjamin Peterson

---

## [Author Response · Author response to Decision Letter 0]

14 Sep 2023

Please include captions for your Supporting Information files at the end of your manuscript, and update any in-text citations to match accordingly. Please see our Supporting Information guidelines for more information: http://journals.plos.org/plosone/s/supporting-information. 

Response: The manuscript and supporting information files have been updated. 

Additional Editor Comments:

Dear authors

Thank you for the effort put into this study. I really enjoyed reading the article and read the results part with great interest. I believe that the work put forward here is relevant and important.

It has been a challenge securing external peer review, which is why only a single external reviewer has commented on the manuscript. Since the journal policy is at least two external peer-reviews, I have also provided extensive reviews and comments to accommodate for this.

Please excuse the sometimes straightforward communication, no offense is meant.

Introduction

1. Regarding references 1-3, consider changing these references or including this PMID: 27939265.

Regarding references 4,5, and 6. Consider the terminology and use of the references. Reference 4 is specific to differences in risk between genders, not just high rates of injury. Reference 5 reports both incidence and prevalence and includes region-specific injury proportions. Consider the relevance of reference 6 since it´s an original study reporting a single identified injury proportion from a cohort of runners.

Furthermore, reference 6 is included in reference 5, and reference 5 is a literature study reporting injury proportion across a range of studies. Therefore, it may be a more accurate estimate of the frequency of injuries. It could also be relevant to include the following reference PMID:25951917 maybe. In that way, you could very nicely describe the nature of RRI regarding frequency, gender differences, performance level differences, and differences between anatomical regions.

Response: References 1-3 have been replaced with the suggested reference (PMID: 27939265). References 4-6 have been updated, and the reported prevalence rate has been changed to a more accurate incidence rate, including a statistic for running-related injuries in recreational runners, as per your suggested reference.

The text used to read: “Despite recreational running being an extremely popular physical activity and sport (1–3), it is associated with high rates of injury (4,5), with prevalence rates as high as 66% (6)”.

The text now reads (lines 41-43): “Despite recreational running being an extremely popular physical activity and sport [1] it is associated with high rates of injury [2], with incidence rates of 40% [3] or 7.7 injuries/1000 hours [4] reported”. 

2. Regarding reference 7, I´m not able to see how this reference supports the notion that most RRI are overuse injuries. I agree that the vast majority of injuries, indeed, are overuse injuries, I just think the reference are inappropriate.

Response: This reference has been updated.

The text used to read: “The vast majority of running-related injuries (RRIs) are overuse injuries (7) which occur when excessive, repetitive loads are applied to tissues beyond their adaptive capability (8–10)”.

The text now reads (lines 43-45): “The vast majority of running-related injuries (RRIs) are overuse injuries [5] which occur when excessive, repetitive loads are applied to tissues beyond their adaptive capability [5–7].

3. Regarding Figure 1 and the introduction part lines 74-89. The thoughts presented here are very theoretical and somehow disrupt the flow of an otherwise coherent introduction. In the discussion, this subject is also included. Consider removing it from the introduction.

Response: We appreciate this comment and agree that the example provided is theoretical and that the introduction reads more cohesively without it. We have removed the figure and associated text from the introduction. 

The text used to read: “It is also important for future research aiming to identify RRI risk factors to understand how lower levels of injury may interact with other risk factors to develop into a significant injury (i.e., as per the consensus definition (19)) (20), or indeed, how these lower levels of injury may themselves be risk factors for injury (20). An example of such a scenario may be as follows (Fig 1). At the time of a runner’s initial biomechanical assessment, they are observed to land with a highly extended knee. Subsequently, this results in a lower level injury (‘complaint’) in their knee; however, they then go on to develop a significant injury in their Achilles tendon. Without having the knowledge of the lower level injury at the knee, the researcher/clinician may have concluded that the extended knee was directly causative of the Achilles injury, and therefore falsely identify it as a risk factor for Achilles tendon injuries. In reality, the lower level injury at the knee may have resulted in the runner changing their running technique (e.g. adopting a more forefoot running technique to reduce loading on the knee) which placed greater loading on the Achilles tendon (27,28), leading to injury. By monitoring the presence of the lower level injury at the knee, the researcher/clinician would better understand how this may have changed the runner’s technique, which was actually the true risk factor for the Achilles injury. In addition, with the dynamic relationship between multiple risk factors and the onset of injury, it is also important to understand how runners react to this process of injury development (i.e. a lower level injury), and how they manage all levels of injury”

The text now reads (lines 73-79): “It is also important for future research aiming to identify RRI risk factors to understand how lower levels of injury may interact with other risk factors to develop into a significant injury (i.e., as per the consensus definition [16]) [17], or indeed, how these lower levels of injury may themselves be risk factors for injury [17]. In addition, with the dynamic relationship between multiple risk factors and the onset of injury, it is important to understand how runners react to this process of injury development (i.e. a lower level injury), and how they manage all levels of injury. This may not only help to differentiate between various levels of injury, but also provide insight into how these levels act as potential risk factors themselves for further injury”.

Methods

1. Several changes need to be made in the design section. First, the applied methodology should be described in the beginning. Secondly, the method of data acquisition could be described.

Much more effort needs to be directed toward describing how the data acquisition method fits the methodological framework. Several of the referenced papers in this section elaborates upon exactly this issue, but the description seems superficial and standardized.

Response: In response to the two comments above, section 3.1 ‘Design’ has been reviewed and updated as per your recommendations. We have detailed the applied methodology earlier in the section, and described how our methods of data acquisition are congruent with the methodological framework.

The text used to read: “Focus groups were deemed an appropriate method of data collection as they can yield rich, in-depth data through the interaction of participants (29–31), and can enhance personal accounts by benefitting from the rapport built in a homogeneous sample with shared experiences (32). Phenomenology was deemed a suitable methodological approach as it focuses on the lived experiences of humans, eliciting insightful accounts of individuals’ subjective experiences regarding a certain topic (33,34). Although previously thought to be unsuitable in phenomenological research (35,36), focus groups can provide greater understanding of the phenomenon of interest (37) as a well-facilitated focus group can produce greater spontaneity, curiosity, and openness to new perspectives on familiar experiences (37–39). Interpretative phenomenology suggests that researchers cannot completely remove themselves from the studied phenomenon (40) and considering how their own biases may affect their interpretation (41), researchers should aim to explain the lived experiences of participants, rather than solely convey them (41,42). A semi-structured focus group schedule was developed during several brainstorming meetings between researchers (AL, EW, SOK, and KM [Supplementary material A]). Question content, sequencing, phrasing and timing were discussed during meetings. The schedule was tested on colleagues to determine its appropriateness, and then used in the pilot study (details below). Ethical approval was granted by the local university’s Ethics Committee. The Standards for Reporting Qualitative Research was adhered to (43) (Supplementary material B)”.

The text now reads (lines 88-105): “Interpretative phenomenology (IP) was deemed to be an appropriate methodological approach because, as a branch of phenomenology, it focuses on the lived experiences of humans, eliciting insightful accounts of individuals’ subjective experiences regarding a certain topic [24,25]. Focus groups were deemed an appropriate method of data collection as they can yield rich, in-depth data through the interaction of participants [26–28], and can enhance personal accounts by benefitting from the rapport built in a homogeneous sample [29]. Although suggested to be incompatible by some [30], executing focus groups with an interpretative phenomenological approach (IPA) was congruent with the aim of the study and allowed for enrichment of the data regarding the phenomenon of interest through the interaction between participants [31]. Despite the complex interactional environment that is created by conducting focus groups, the opportunity to engage with a homogenous sample (i.e., runners) as a group can elicit insightful and experiential data [32]. While it has been suggested that there is difficulty in developing phenomenological accounts of data due to the complexity of group dynamics (from interactional, social and contextual perspectives) [33], the complexity of individual and shared contexts [33], and the influence and position of the researcher(s) [33–35], the free-flowing and engaging nature of focus group discussion can allow for social interaction of shared, similar or conflicting lived experiences of the phenomenon of interest [31,36].

2. The choice of focus groups is not a problem. However, no consideration is given to how the interaction between the group members is handled. This is especially relevant in terms of analysis, and from what I can understand, video recordings of the interviews exist. Both reference 35 and 37 provide insight and discussion surrounding this issue.

Response: Many thanks for your comment. We understand that a complex interactional environment is created by executing focus groups, however, with interpretative phenomenology, the aim is not to achieve consensus on any particular topic, but rather the exploration of participants’ lived experience. The interaction generated between participants allowed us to capture a more detailed account of these lived experiences as it allowed for social interaction on shared, similar or conflicting experiences (Gaskell and Willaims, 2018). Additionally, the video recordings of the focus groups were only used to confirm speaker identity, correct any discrepancies in the transcripts, and add further nuance/detail to the transcripts, such as gestures towards the white board during the drawing exercises. 

The text now reads (lines 97-105): “Despite the complex interactional environment that is created by conducting focus groups, the opportunity to engage with a naturally occurring sample (i.e., runners) as a group can elicit insightful and experiential data [32]. While it has been suggested that there is difficulty in developing phenomenological accounts of data due to the complexity of group dynamics (from interactional, social and contextual perspectives) [33], the complexity of individual and shared contexts [33], and the influence and position of the researcher(s) [33–35], the free-flowing and engaging nature of focus group discussion can allow for social interaction of shared, similar or conflicting lived experiences of the phenomenon of interest [31,36]”. 

3. Regarding reference 36, I´m not able to see how this reference supports the notion that focus groups are unsuitable for phenomenological research.

Response: Apologies, this was an error and the reference has been removed.

4. Please be explicit and precise about the choice of methodology instead of only elaborating slightly upon interpretative phenomenology.

Response: Elaboration has been provided on our choice of Interpretative Phenomenology as a methodology, with further detail and references being provided. 

The text used to read: “Focus groups were deemed an appropriate method of data collection as they can yield rich, in-depth data through the interaction of participants (29–31), and can enhance personal accounts by benefitting from the rapport built in a homogeneous sample with shared experiences (32). Phenomenology was deemed a suitable methodological approach as it focuses on the lived experiences of humans, eliciting insightful accounts of individuals’ subjective experiences regarding a certain topic (33,34). Although previously thought to be unsuitable in phenomenological research (35,36), focus groups can provide greater understanding of the phenomenon of interest (37) as a well-facilitated focus group can produce greater spontaneity, curiosity, and openness to new perspectives on familiar experiences (37–39). Interpretative phenomenology suggests that researchers cannot completely remove themselves from the studied phenomenon (40) and considering how their own biases may affect their interpretation (41), researchers should aim to explain the lived experiences of participants, rather than solely convey them (41,42). 

The text now reads (lines 88-105): “Interpretative phenomenology (IP) was deemed to be an appropriate methodological approach because, as a branch of phenomenology, it focuses on the lived experiences of humans, eliciting insightful accounts of individuals’ subjective experiences regarding a certain topic [24,25]. Focus groups were deemed an appropriate method of data collection as they can yield rich, in-depth data through the interaction of participants [26–28], and can enhance personal accounts by benefitting from the rapport built in a homogeneous sample [29]. Although suggested to be incompatible by some [30], executing focus groups with an interpretative phenomenological approach (IPA) was congruent with the aim of the study and allowed for enrichment of the data regarding the phenomenon of interest through the interaction between participants [31]. Despite the complex interactional environment that is created by conducting focus groups, the opportunity to engage with a naturally occurring sample (i.e., runners) as a group can elicit insightful and experiential data [32]. While it has been suggested that there is difficulty in developing phenomenological accounts of data due to the complexity of group dynamics (from interactional, social and contextual perspectives) [33], the complexity of individual and shared contexts [33], and the influence and position of the researcher(s) [33–35], the free-flowing and engaging nature of focus group discussion can allow for social interaction of shared, similar or conflicting lived experiences of the phenomenon of interest [31,36]”. 

5. Lines 117-119 should be moved to study procedures.

Response: This section has been moved to the section ‘Pilot study’ (lines 117-127)

6. Lines 119-121, starting with “The schedule……..” should be moved to pilot study or removed.

Response: These lines have been moved to the section ‘Pilot study’ (lines 119-120).

7. Lines 132-133 describe that all participants reported a lower level injury. However, in the result section lines 253-256, it is described that not all participants have experienced a RRI. I assume it is because a distinction between a RRI and a lower level injury is made. But, chronologically, the reader is not familiar with a lower level injury based on the developed running injury continuum. Therefore, this may be more appropriate elsewhere, maybe in the result section….

Response: Yes this is due to the distinction between lower level injuries and running-related injuries. We reported this finding originally in section 3.2 Participants (of the Methods section) as we felt it was appropriately related to participants’ lived experience, but on reflection, we concur that it is more suitable to be stated in the Results. This statement has been moved to the section ‘Demographics’ (line 244).

8. Please provide further information on the participants in the individual focus groups. And elaborate on if the 2 groups used in the pilot study are included in the total sample of 7 groups.

Response: Only brief details (i.e., gender and age) regarding participants of the pilot focus groups were collected on the group. It has also been clarified that the data obtained in the pilot study were not included in the main results.

The text used to read: “Five male and five female participants were recruited as a convenience sample, aged 23.8 ± 5.9 years. Two focus groups were moderated by two researchers (AL and SOK), each taking place in-person, on University grounds, and lasted 59.7 ± 5.6 minutes. Focus groups were audio and video recorded, and transcribed verbatim”

The text now reads (line 122-127): “Five male and five female physically active participants were recruited as a convenience sample, aged 23.8 ± 5.9 years. Two focus groups were moderated by two researchers (AL and SOK), each taking place in-person, on University grounds, and lasted 59.7 ± 5.6 minutes. Focus groups were audio and video recorded, and transcribed verbatim. Data obtained from the pilot study are not included as part of the results”.

9. Line 191-193 mentions the development and expansion of themes. This warrants further elaboration. Consider including supplementary material C in the article instead of Figure 2. Because supplementary material C provides valuable insight into the data analysis process, and Figure 2 is very superficial. Further, supplementary material C is mentioned several times in the manuscript, which further highlights its relevance.

Response: We have elaborated on this matter and provided explanation of how we used the ‘order of themes’ document during data collection and analysis. We originally did not include Supplementary Material C in the main text as it is quite an extensive table and we felt that it may detract from the main findings; however, taking your recommendations on board, we have included it in the main text (Tables 2 and 3 in the Results section of the manuscript). However, if you feel that the extensiveness of the tables (particularly Table 2) breaks up the presentation of the results and detracts from the richness of the findings (i.e., participant quotations and the Running Injury Continuum figure [Figure 1]), we will be happy to move the tables into the supplementary material. Additionally, we have moved Figure 2 to the supplementary material (S1 Fig). 

The text used to read: “An ‘order of themes’ document was developed during the pilot phase, and was reflexively updated based on the developing, merging and expanding themes and sub-themes in order to organise the codes, sub-themes, themes and core categories (Supplementary material C)”. 

The text now reads (line 181-184): “The ‘order of themes’ document was used to organise the codes, sub-themes, themes and core categories and was reflexively updated throughout data collection and analysis phases. Based on the developing coding, further levels of sub-themes were developed while some sub-themes were merged”.

Results

1. Table 1, is very hard to read. Please consider rearranging the table. I would suggest having the headlines as columns and the categories as rows.

Response: Table 1 has been re-formatted to consist of just 2 columns. 

Table 1 used to read: 

Table 1: Running practices and injury history 

Preferred running event (n=31)*

<5km n=5 (16%) Marathon n=3 (10%) 

5km n=21 (67%) Ultramarathon n=1 (3%) 

10km n=18 (58%) Triathlon n=1 (3%)

16km n=4 (13%) Other n=1 (3%)

Half marathon n=6 (19%) 

Weekly mileage (n=31) 

<10km/week n=1 (3%) 31-40km/week n=1 (3%)

10-20km/week n=12 (39%) 41-50km/week n=5 (16%) 

21-30km/week n=10 (32%) >50km/week n=2 (7%) 

Amount of missed training with worst ever RRI (n=27) 

<1 week n=2 (7%) 4-6 weeks n=2 (7%) 

7-10 days n=4 (15%) >6 weeks n=13 (48%) 

2-3 weeks n=5 (19%) Unsure n=1 (4%) 

Table 1 now reads as (line 247):

Table 1: Running practices and injury history

Preferred running event (n=31)* <5km: 16% (n=5)

 5km: 67% (n=21)

 10km: 58% (n=18)

 16km: 13% (n=4)

 Half marathon (21.1km): 19% (n=6)

 Marathon (42.2km): 10% (n=3)

 Ultramarathon (>42.2km): 3% (n=1)

 Triathlon: 3% (n=1)

 Other: 3% (n=1)

Weekly mileage (n=31) <10km: 3% (n=1)

 10-20km: 39% (n=12)

 21-30km: 32% (n=10)

 31-40km: 3% (n=1)

 41-50km: 16% (n=5)

 >50km: 7% (n=2)

Amount of missed training with worse ever RRI (n=27) <1 week: 7% (n=2)

 7-10 days: 15% (n=4)

 2-3 weeks: 19% (n=5)

 4-6 weeks: 7% (n=2)

 >6 weeks: 48% (n=13)

*: multiple choice available, RRI: running-related injury

2. Consider changing athletic therapist to physiotherapist in the figure, since all included narratives in the manuscript mentions physiotherapists.

Response: We have added ‘Physiotherapist’ to the figure, but have kept ‘Athletic Therapist’ as some participants specified ‘Athletic Therapist’ during focus groups. In addition, Athletic Therapists are common healthcare professionals in Ireland and we feel that it is reflective of the data collected and the sample population. 

The figure used to read:

The figure now reads (line 260):

Discussion

1. More precision is needed in discussing how and where the RRI consensus definition overlaps with the presented running injury continuum. Especially considering the perspective put forward in relation to both clinical practice and future research.

Response: We have elaborated on how higher level injuries overlap with the consensus definition in terms of runners’ description of higher level injuries and the criteria used in the consensus definition. 

The text used to read: “Lower level injuries span between ‘discomfort’ to ‘non-responding niggle’, while higher level injuries, which are most associated with the RRI consensus definition (19), span between ‘injury: short-term effect’ to ‘career-ending injury’”.

The text now reads (line 548-573): “Lower level injuries span between ‘discomfort’ to ‘non-responding niggle’, while higher level injuries, which are most associated with the RRI consensus definition [16], span between ‘short-term injury’ to ‘career-ending injury’. The consensus definition defines a RRI as: “running-related (training or competition) musculoskeletal pain in the lower limbs that causes a restriction on or stoppage of running (distance, speed, duration, or training) for at least 7 days or 3 consecutive scheduled sessions, or that requires the runner to consult a physician or other healthcare professional” [16, p.377]. Higher level injuries overlap with this definition by virtue of the commonalities between our participants’ description and the criteria used in the consensus definition. Firstly, a description of pain is associated with higher level injuries and is required to define injury within the consensus definition [16]. Although pain is mentioned in earlier levels of the Running Injury Continuum, it is described in milder forms and becomes significantly more prominent at these higher level injuries. Secondly, within higher level injuries, runners will stop running, at least in the short-term (e.g., a week), a criterion and time-frame strongly associated with the consensus definition [16]. While earlier lower levels of injury were described as causing restrictions to running (such as reducing load), they were not associated with this length of a time-frame, distinguishing them from the consensus definition. Finally, higher level injuries result in runners requiring medical attention, an evident criterion in the consensus definition [16]. It can be argued that descriptions of the ‘non-responding niggle’ (medium pain causing alterations to training, short-term rest, or HCP intervention) are consistent with elements of the consensus definition; however, this level is not included as a higher level injury for two reasons. Firstly, the opinions of participants varied across the categories of descriptors (e.g., some described continued running, whereas others described a stoppage to running), with more agreement being evident within higher level injuries. Secondly, there was a strong description of this level being a ‘crossroads’, more so as a transition level from lower to higher level injuries”.

2. When discussing the education of runners from a preventive standpoint, please include and discuss some of the available knowledge on the effect of educational interventions to reduce the risk of RRI, relative to the running injury continuum.

Response: We have added text discussing the use of education in injury prevention interventions, and how this relates to the Running Injury Continuum. 

This text reads (lines 708-722): Support for the relationship between education and injury prevention (IP) has been described in terms of the translation of knowledge to enhance the adoption of IP interventions [61], and enhancing the recovery process to prevent injuries [62]. Research specifically examining this relationship involving runners seems to be limited; however, an online IP intervention consisting of educational videos informing participants about the aetiology and mechanisms of injury, combined with evidence-based IP advice was shown to have a positive effect on knowledge, attitude, intention, and behaviour, immediately and three months following the intervention [63]. Runners’ perception of injury risk and their attitudes towards the importance of executing IP measures were positively affected by the intervention [63]. Furthermore, another study examining the effectiveness of an online IP intervention found no significant effect of their intervention on actual preventative behaviour [60]. One suggestion for this was a difference in the content of the IP interventions, such as the educational videos which were included in Adriaensens and colleagues’ [63] study [60]. These findings highlight that with enhanced knowledge and education regarding injury risk and management, runners are more likely to adopt IP practices.

3. The validity of the running injury continuum needs thorough discussion. The sample size is highlighted, but data saturation is not discussed. Why is that? The following reference may provide useful information for the discussion PMID: 25951917. 

A discussion/sharing of experiences about how the interpretative phenomenological approach and using focus groups as a method of collecting data is warranted.

Response: Many thanks for your comments. Rather than specifically discussing validity, we present multiple methods of trustworthiness and data triangulation in the Methods section as means of enhancing the rigour of our findings (lines 198-227). We do not speak to data saturation as it is generally not suitable with IPA (Brocki and Wearden, 2006). Instead, our sample size allowed us to address the aims of our study, explore its broad scope, and capture richness and depth of the data. Text has been added to the discussion highlighting the use of IPA and focus groups to capture this richness of data. Additionally, a comment has been added to the beginning of the discussion referring to the richness and depth of data that was collected through the execution of focus groups, using an Interpretative phenomenological approach. 

This text reads (line 541-543): “The current study used an IPA to explore this topic, and the authors cannot overstate the richness and depth of data that was captured, primarily facilitated by the social interaction between participants during focus groups”.

Reviewers' comments:

Reviewer's Responses to Questions

Comments to the Author

1. Is the manuscript technically sound, and do the data support the conclusions?

Reviewer #1: Yes

2. Has the statistical analysis been performed appropriately and rigorously?

Reviewer #1: Yes

3. Have the authors made all data underlying the findings in their manuscript fully available?

Reviewer #1: Yes

4. Is the manuscript presented in an intelligible fashion and written in standard English?

Reviewer #1: Yes

5. Review Comments to the Author

Reviewer #1: I thank the editor for the opportunity to review this manuscript. This study explored the perspectives of runners regarding running-related injury and their management strategies when managing injury. This qualitative study is well-justified and contributes to filling a gap in running-related injury research where further qualitative research is needed. The authors should be commended for undertaking this work with quite a large sample size, which can involve substantial work in the context of the thematic analysis method. Additionally, I am pleased to see that this study explored perspectives of both male and female sexed runners.

A novel aspect of this study is the description of a ‘running injury continuum’ which is an astute observation and interpretation of the available data. In particular, this continuum represents well both the experiential aspects of RRI (of different levels) but also links these individual (but overlapping) levels and the kinds of health-seeking behaviours which runners may display.

I have several minor comments which are provided in a genuine attempt to improve aspects of this manuscript:

1. First, the authors have done great work describing a new injury continuum and a new theoretical model for injury development. However, the continuum depicted in figure 1 seems a bit simplistic, given the multifactorial aetiology of RRI. I wonder if this model could be revised to include more preceding factors other than simply a change in running biomechanics (which have a debatable association with RRI aetiology).

Response: Thank you for your thoughts, and we agree that the figure is a simplistic version of describing a potential mechanism for the onset of RRIs. Taking your comments and those from another reviewer into account, we have decided to remove the figure and associated text regarding the example we provided from the introduction. 

The text in the introduction used to read: “It is also important for future research aiming to identify RRI risk factors to understand how lower levels of injury may interact with other risk factors to develop into a significant injury (i.e., as per the consensus definition (19)) (20), or indeed, how these lower levels of injury may themselves be risk factors for injury (20). An example of such a scenario may be as follows (Fig 1). At the time of a runner’s initial biomechanical assessment, they are observed to land with a highly extended knee. Subsequently, this results in a lower level injury (‘complaint’) in their knee; however, they then go on to develop a significant injury in their Achilles tendon. Without having the knowledge of the lower level injury at the knee, the researcher/clinician may have concluded that the extended knee was directly causative of the Achilles injury, and therefore falsely identify it as a risk factor for Achilles tendon injuries. In reality, the lower level injury at the knee may have resulted in the runner changing their running technique (e.g. adopting a more forefoot running technique to reduce loading on the knee) which placed greater loading on the Achilles tendon (27,28), leading to injury. By monitoring the presence of the lower level injury at the knee, the researcher/clinician would better understand how this may have changed the runner’s technique, which was actually the true risk factor for the Achilles injury. In addition, with the dynamic relationship between multiple risk factors and the onset of injury, it is also important to understand how runners react to this process of injury development (i.e. a lower level injury), and how they manage all levels of injury”

The text now reads (lines 73-81): ““It is also important for future research aiming to identify RRI risk factors to understand how lower levels of injury may interact with other risk factors to develop into a significant injury (i.e., as per the consensus definition [16]) [17], or indeed, how these lower levels of injury may themselves be risk factors for injury [17]. In addition, with the dynamic relationship between multiple risk factors and the onset of injury, it is important to understand how runners react to this process of injury development (i.e. a lower level injury), and how they manage all levels of injury. This may not only help to differentiate between various levels of injury, but also provide insight into how these levels act as potential risk factors themselves for further injury”.

2. Introduction line 65-68: “and the identification of a ‘complaint’ level in the process of injury development (prior to time-loss, training restriction or seeking medical attention) (22).” This sentence feels incomplete or may require grammatical revision.

Response: This sentence has been re-phrased.

The text used to read: “These studies reported that RRIs are perceived as progressive, with injury ‘categories’ suggested in one, although not described further (25), and the identification of a ‘complaint’ level in the process of injury development (prior to time-loss, training restriction or seeking medical attention) (22)”. 

The text now reads (line 64-68): “These studies reported that RRIs are perceived as progressive, with injury ‘categories’ suggested in one study, although not described further [22], and the identification of a ‘complaint’ level in the process of injury development (prior to time-loss, training restriction or seeking medical attention) described in another [19]”.

3. Introduction line 72: perhaps ‘gradual and multifactorial’ would be better phrased as ‘multifactorial and progressive’, at the authors’ discretion.

Response: This has been updated.

The text used to read: “Therefore, a greater understanding of this process of injury development (i.e., runners’ description and management of each level of this process) is clearly needed if researchers and clinicians are to better understand the gradual and multifactorial nature of RRIs, their risk factors, and ultimate prevention”

The text now reads (line: 69-72): “Therefore, a greater understanding of this process of injury development (i.e., runners’ description and management of each level of this process) is clearly needed if researchers and clinicians are to better understand the multifactorial and progressive nature of RRIs, their risk factors, and ultimate prevention”

4. Methods/design line 121-122: is there an ethics approval number which should be listed?

Response: The ethics reference number has been added (line 131).

5. Line 122 “were adhered to”

Response: This has been updated.

The text used to read: “The Standards for Reporting Qualitative Research was adhered to (43) (Supplementary material B)”. 

The text now reads (line 132): “and the Standards for Reporting Qualitative Research were adhered to [38] (S2 Table)”.

6. Participants, line 132-133: can the authors describe how they confirmed all participants had experienced a ‘lower level injury’?

Response: Runners answered questions relating to their experience of running-related pain/discomfort, indicating lower level injuries in a hard-copy questionnaire completed after their focus group. This has been clarified in the Methods and Results sections.

The text used to read: “All participants reported previously experiencing a lower level injury”. 

The text now reads: “Participants were asked to complete a short individual questionnaire (hard-copy) gathering further demographic information, training practices, and injury history (including their experience of running-related pain/ discomfort)” (line 148-150).

Additionally, the line “All participants reported previously experiencing a lower level injury” has been moved to the Results section (line 244). 

7. I apologise if I have missed this somewhere throughout the manuscript, but could the authors clarify how the participant demographic information was captured? Was this through a paper-based survey? Are all survey data presented in the manuscript? Please also include the questioning methods e.g. Likert scales.

Response: Apologies for not clarifying. Demographic data were collected via hard-copy questionnaire at the end of each focus group (clarified in line 149). Not all survey data are presented, but all data relevant to this study are presented. 

The text now reads (line 149-150): “Participants were asked to complete a short individual questionnaire (hard-copy) gathering further demographic information, training practices, and injury history (including their experience of running-related pain/ discomfort)”.

8. Whilst reading the results, I quickly identified that there is overlap between individual levels of injury with respect to self-management, peer support, and health-seeking behaviours, and this is very well addressed by the authors in the discussion.

Response: Many thanks for this comment. Yes, we feel this is an important finding and reflects the progressive development of RRIs. 

9. Completely discretionary comment, but I wonder if the terms ‘short term injury’ and ‘longer term injury’ would be more easily adopted than ‘injury: short-term effect/long-term effect’?

Response: Thank you for your thoughts. We have given the comment some consideration and feel that including ‘effect’ does not add a significant amount of description to the terms, and feel your suggestion of ‘short-term injury’ and ‘long-term injury’ may be more easily adopted. This has been changed throughout the manuscript and relevant figures. 

10. Career ending injury – had any of the runners experienced a ‘career ending injury’ which caused permanent cessation of running?

Response: Taking data from the discussions in the focus groups and the hard-copy questionnaires that participants completed, no runners had experienced a career-ending injury, however, this was their perception of the ‘worst possible injury’ a runner could sustain (i.e., one that would result in them never being able to run again), and a suitable end point for the development of injury process. A line clarifying this has been added to the results.

The text now reads (line 483-485): “While no participant reported experiencing a ‘career-ending injury’, this was the perceived as the most severe injury a runner could sustain, and a suitable end-point for the Running Injury Continuum”. 

11. Line 564 “not individually unique” suggest alternate phrasing as ‘individual’ and ‘unique’ sound quite synonymous

Response: The word “individually” has been removed.

The text used to read: “While the categories of descriptors used to differentiate levels of injury were sometimes not individually unique (e.g., caution was used to psychologically describe both ‘niggle’ and ‘twinge’)…” 

The text now reads (line 589): “While the categories of descriptors used to differentiate levels of injury were sometimes not unique (e.g., caution was used to psychologically describe both ‘niggle’ and ‘twinge’)…”

References

Brocki, J.M. and Wearden, A.J. (2006) “A critical evaluation of the use of interpretative phenomenological analysis (IPA) in health psychology, Psychology & Health, 21(1), pp. 87-108. 

Gaskell, L. and Williams, A.E. (2018) “A qualitative study of the experiences and perceptions of adults with chronic musculoskeletal conditions following a 12-week Pilates exercise programme”, Musculoskeletal Care, pp. 1-9

---

## [Editor Report · Decision Letter 1]

19 Sep 2023

The Running Injury Continuum: A qualitative examination of recreational runners’ description and management of injury

PONE-D-23-10426R1

Dear Dr.Lacey

We’re pleased to inform you that your manuscript has been judged scientifically suitable for publication and will be formally accepted for publication once it meets all outstanding technical requirements.

Kind regards,

Daniel Ramskov, Ph.D

Academic Editor

PLOS ONE
---

## [Editor Report · Acceptance letter]

25 Sep 2023

PONE-D-23-10426R1 

The Running Injury Continuum: A qualitative examination of recreational runners’ description and management of injury 

Dear Dr. Lacey:

I'm pleased to inform you that your manuscript has been deemed suitable for publication in PLOS ONE. Congratulations! Your manuscript is now with our production department. 

Kind regards, 

on behalf of

Dr. Daniel Ramskov 

Academic Editor

PLOS ONE